# Photo-modulated activation of organic bases enabling microencapsulation and on-demand reactivity

Wenle Li [1,5] ✉, Xiaocun Lu[2,5], Jacob M. Diamond [3], Chengtian Shen[4], Bo Jiang[1], Shi Sun[1], Jeffrey S. Moore [4] & Nancy R. Sottos [3] ✉

A method is developed for facile encapsulation of reactive organic bases with potential application for autonomous damage detection and self-healing polymers. Highly reactive chemicals such as bases and acids are challenging to encapsulate by traditional oil-water emulsion techniques due to unfavorable physical and chemical interactions. In this work, reactivity of the bases is temporarily masked with photo-removable protecting groups, and the resulting inactive payloads are encapsulated via an in situ emulsion-templated interfacial polymerization method. The encapsulated payloads are then activated to restore the organic bases via photo irradiation, either before or after being released from the core-shell carriers. The efficacy of the photo-activated capsules is demonstrated by a damage-triggered, pH-induced color change in polymeric coatings and by recovery of adhesive strength of a damaged interface. Given the wide range of potential photo-deprotection chemistries, this encapsulation scheme provides a simple but powerful method for storage and targeted delivery of a broad variety of reactive chemicals, promoting design of diverse autonomous functionalities in polymeric materials.

Living organisms exhibit remarkable self-adaptive properties, effortlessly transporting vital biochemical substances and adeptly responding to changes in their surroundings. In the realm of synthetic materials, the quest for adaptive characteristics has led to the encapsulation of functional fluids within core-shell microstructures[1–7]. These microscale capsules have been harnessed within engineered materials to craft responsive polymers[8–16], employed as intelligent carriers for precision drug delivery[17], utilized as energy-efficient sorbents for the reversible capture and release of carbon dioxide[18], and seamlessly integrated into transient electronics, enabling on-demand device disintegration[19]. Generally, liquid cores consist of highly reactive compounds such as bases and acids are preferred payloads because of their abilities to activate diverse functions with minimum capsule

doses. For example, polymer-walled microcapsules containing organic bases were embedded in protective coatings with separately encapsulated dyes to devise mechanochromic responses[20,21]. Upon mechanical rupture, the damage-released bases caused deprotonation of the pH sensitive dyes, generating a local color change to autonomously report crack propagation. When the basic payloads were composed of polyfunctional amines, they became capable of initiating an in situ self-healing function to restore mechanical properties by crosslinking epoxy resins that were simultaneously released from a secondary microcontainer[22,23].

Although organic bases have been encapsulated previously, the final capsules often exhibit undesirable properties such as less controlled structure, weak shell wall, and poor stability. Li et al. reported

[1]School of Materials Science and Engineering, China University of Petroleum (East China), Qingdao, Shandong 266580, China. [2]Department of Chemistry and Biomolecular Science, Clarkson University, Potsdam, NY 13699, USA. [3]Department of Materials Science and Engineering, Beckman Institute for Advanced Science and Technology, University of Illinois at Urbana-Champaign, Urbana, IL 61801, USA. [4]Department of Chemistry, Beckman Institute for Advanced Science and Technology, University of Illinois at Urbana-Champaign, Urbana, IL 61801, USA. [5]These authors contributed equally: Wenle Li, Xiaocun Lu. ✉e-mail: wli@upc.edu.cn; n-sottos@illinois.edu

a Pickering emulsion-templated method to prepare aliphatic amine microcapsules[24]. Hydrophobic clay nanoplatelets were employed to promote partitioning of the hydrophilic payloads in the continuous oil phase and stabilize the emulsion droplets. Capsules were created by the sequential interfacial polymerization between isocyanates and amines. Due to the dual role of amines as payload and shell former[25], this technique illustrated a fundamental challenge in developing capsules with well-defined core-shell structures. In order to reduce the unpreferable consumption of payloads, McIlroy et al. introduced a highly reactive crosslinker in a similar inverse Pickering emulsion system to maximally constrain amines in the liquid cores[26]. Nevertheless, the resulting capsules showed non-spherical morphologies with a core-shell ratio less than 50%. Thereafter, a nonaqueous emulsion approach was investigated to further optimize amine distribution. Lu et al. designed an oil-in-oil Pickering emulsion, in which the amine payload was restricted in the dispersed phase with the assistance of a partitioning inhibitor[27]. While the emulsion stability was improved, core-shell structure and mechanical properties of the capsules were not revealed in this study. With another attempt to minimize amine participation in the shell-forming reactions, Zhang et al. coupled the above synthesis solution with a microfluidic device[28]. Individual micro-droplets were deposited in the reaction solution, leading to a rapid encapsulation via interfacial polymerization before an equilibrium emulsion was formed. Capsules with a polyamine payload were successfully prepared, but the polyurea shell wall was deformed due to shrinkage during the rapid shell formation process. Chen et al. modified the microfluidic approach, formulating a double-emulsion template that was subsequently consolidated by a UV-induced polymerization[29]. The controlled laminar flow circumvented the miscibility issues and monodisperse microcapsules filled with reactive amines were fabricated. However, due to the sequential production of micro-droplets, the throughput of this technique remains limited, presenting challenges for scaling up production. Another prominent scheme to encapsulate organic bases exploited a phase separation process. Li et al. dissolved poly(methyl methacrylate) along with amines in an organic solvent and emulsified the oil phase in an aqueous solution[30]. Safdari et al. atomized a mixture containing dispersed poly(styrene co-acrylonitrile), amines, and solvent via the electrospray technology[31]. In both studies, polymeric membranes formed around amine droplets as they phased out of the emulsions during solvent evaporation. Because the shell walls were framed with non-crosslinked polymers in a less controlled manner, the prepared microcapsules retained porous shell membranes, less defined geometries, and low amine contents. To avoid intervention of the highly reactive payloads on their microencapsulation, a two-step approach was developed to load organic bases into prefabricated core-shell structures. Under vacuum assistance, a variety of polyfunctional amines were infiltrated into hollow polymeric microspheres[32], hollow carbon nanospheres[33], or etched glass bubbles[34,35]. Despite the high amine loading achieved, these capsules were unstable and leaked payloads over time through the microscopic holes in the shell membranes. Moreover, excessive amines adhering to the outer surfaces of the capsules were inescapable due to the infiltration process, suggesting a poor encapsulation. On top of the various issues of the above technologies, environmental stabilities of current amine microcapsules are suspicious as oxygen will inevitably penetrate through the shell materials and consume the active content gradually.

Herein, we describe a scheme for facile encapsulation of organic bases and demonstrate mechanically triggered release and reactive response in a polymer. The concept frames a photo-activation mechanism in an emulsification polymerization system, configuring a reactivity trigger for the payload to selectively promote emulsion stability and release of actives. The reactivities of three types of aliphatic amines are successfully masked with a photo-removable protecting group. Each inactive payload is then encapsulated with highly defined core-shell microstructures, and the amine functionalities are restored on demand via a controlled irradiation of UV light (Fig. 1). The designed chemical modification renders amines compatible with the shell formers and suppresses their amphiphilic properties, ensuring that the synthesis emulsions are fundamentally stable. By formulating capsules with nanosized shell thickness, this technique grants light transmission through the intact polymer membranes, allowing flexible amine activation either before or after release of the payloads. This capability of regenerating amines on demand eliminates the possible environmental degradation and significantly extends shelf-life of the microcapsules. The activated amine capsules are fully functional to initiate early detection of mechanical damage in polymeric materials and restore adhesion of damaged interfaces. This method, empowered by a wide range of stimuli-triggered deprotection

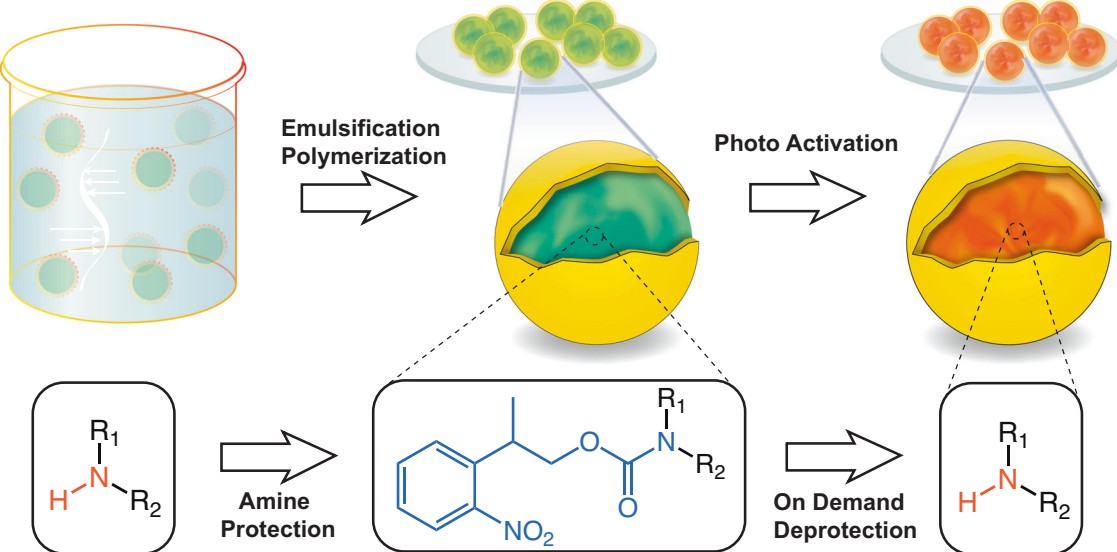

**Fig. 1 | Schematic of microencapsulation approach.** Reactive amines are chemically modified with a photo-removable protection group (i.e. NPPOC) and the prepared inactive payloads are encapsulated via an in situ emulsification polymerization method. Capsules containing photo-responsive payloads are exposed to UV irradiation to activate the reactive ingredients inside the intact shell-walls.

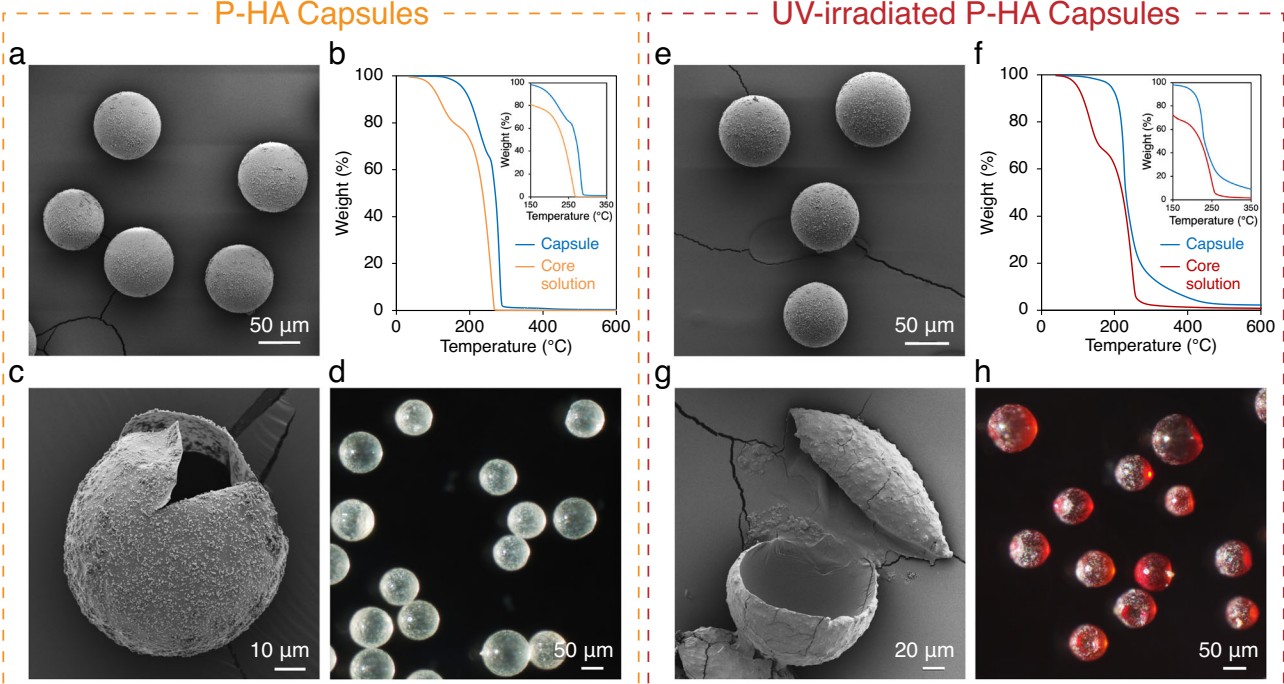

**Fig. 2 | Structural characterization of microcapsules before and after UV irradiation.** SEM images of (**a**) intact and (**c**) ruptured capsules containing NPPOC-protected hexylamine (P-HA) payloads. **b** TGA of P-HA capsules demonstrates significantly enhanced thermal stability compared to the unencapsulated core solution. The inset offers a magnified view of the temperature range from 150 °C to 350 °C. **d** Stereomicrograph of as-prepared P-HA capsules showing a clear, off-white appearance. SEM images of UV-irradiated (**e**) intact and (**g**) ruptured P-HA capsules. **f** TGA of UV-irradiated P-HA capsules verifies the preservation of enhanced thermal stability compared to the unencapsulated, UV-irradiated core solution. The inset offers a magnified view of the temperature range from 150 °C to 350 °C. **h** Stereomicrograph of UV-irradiated P-HA capsules showing a brown color. Each experiment was independently repeated three times, yielding consistent results.

chemistries, opens an avenue for encapsulation of highly reactive chemicals and development of adaptive materials.

## Results

### Microencapsulation of amines enabled by temporary reactivity masking

From the various photo-removable protecting groups that have been previously reported[36], we selected 2-(2-nitrophenyl)propoxycarbonyl (NPPOC) (Fig. 1, see Supplementary Information for details) for this study. In addition to a rapid photo-induced deprotection rate and a relatively small molecular weight[37], NPPOC releases a nitrostyrene derivative upon photolysis, which offers a potentially useful compound to devise self-healing functions via triggered free radical polymerization. NPPOC-protected amines were obtained by a modified synthesis route (Supplementary Fig. 1)[38] and blended with 20 wt% ethyl phenyl acetate (EPA) and 2 wt% *p*-dimethylbenzene (DMB) to afford the capsule payloads. Although direct encapsulation of pure NPPOC-protected amines was achieved, a small addition of EPA as a solvent was found to enhance the photolysis efficiency as well as the payload flowability, both of which facilitated the subsequent delivery of active amines. When added into an aqueous synthesis solution, the inactive payloads formed a stable oil-in-water emulsion. Capsules containing NPPOC-protected amines were therefore fabricated via a modified in situ emulsification condensation polymerization method[39,40], using poly(urea-formaldehyde) as the shell material (see Methods for details).

We first investigated a mono-functional primary amine, hexylamine (HA), to demonstrate this encapsulation concept. NPPOC-protected hexylamine (P-HA) was prepared and its molecular structure was confirmed with [1]H NMR before and after encapsulation. Microcapsules filled with this payload were isolated from the synthesis solution, rinsed with deionized water, and dried in ambient conditions for sequential characterizations. The poly(urea-formaldehyde) shell

wall has a Young's modulus of ~3.7 GPa[41], and was robust enough to survive processing conditions. Scanning electron microscopy (SEM) images show that the obtained capsules were spherical in shape with an average diameter of $93 \pm 12$ μm (Fig. 2a). Successful encapsulation was confirmed through thermogravimetric analysis (TGA) where the solvent evaporation upon dynamic heating was held off until the degradation of the shell structure (Fig. 2b). The capsules exhibited excellent thermal stability with a weight loss less than 5% when heated to 180 °C. Core-shell structure of the capsules was revealed by mechanical rupture using a razor blade and the thickness of shell walls was approximately 100 nm based on analysis of SEM image (Fig. 2c). This thin shell membrane ensures high payload loading and suggests good UV transparency. Stereomicrograph shows that the P-HA capsules were clear with an off-white color (Fig. 2d), further indicating their good light transmission that facilitates the following photo activation of amines.

### Photo activation of amines in intact microcapsules

The prepared P-HA capsules were then exposed to 365 nm UV light to achieve HA restoration. Capsules irradiated at a UV intensity of 30 mW·cm⁻² for 1 h were examined via SEM and TGA. The microcapsules remained intact with no alterations in morphology (Fig. 2e) and no obvious degradation in thermal stability (Fig. 2f). Examination on an intentionally ruptured capsule illustrated the unchanged core-shell structure and showed good fluidity of the payloads as they flowed out of the semi-shell structure (Fig. 2g). The capsules acquired a brown color upon exposure to the UV light (Fig. 2h), which was attributed to the photolysis byproduct 2-(2'-nitrophenyl)propene and the potential intermediates (Supplementary Fig. 2)[42,43]. Payload solutions were extracted from the UV-irradiated P-HA capsules and analyzed with [1]H NMR (Supplementary Fig. 3), providing evidence of the photo-induced removal of the NPPOC protecting group and confirming the well-documented photolysis mechanism (Supplementary Fig. 4).

Additional chemical and optical analyses of the solvent EPA, the internal NMR reference DMB, and control capsules containing such compounds confirmed the stability of these additives throughout the UV exposure (Supplementary Figs. 5–8).

To demonstrate the versatility of this approach, we further encapsulated polyfunctional primary amine 2,2′-(ethylenedioxy)bis(ethylamine) (EA) and secondary amine piperidine (PA). Mechanically robust and thermally stable capsules containing the protected-amines (P-EA and P-PA) were successfully fabricated respectively (Supplementary Fig. 9). On-demand activation of amines was accomplished using the same UV irradiation protocol. The photo-exposed microcapsules retained exceptional structural integrity and thermal resilience, concurrently rejuvenating active amines within the unaltered core-shell structures (Supplementary Fig. 10).

## Photolysis kinetics and controlled amine restoration

The controlled re-establishment of amines within intact capsules was investigated by examining the photolysis kinetics. Microcapsules were exposed to 365 nm UV light at various intensities and the change of payload composition over time was analyzed by $^1$H NMR. Exploiting DMB as an NMR reference, capsules were prepared with payloads containing 2 wt% DMB and the restoration rate of amine was determined by the relative concentration of the activated amine and the initial protected amine (see Supplementary Fig. 11 for a detailed description of the methodology and illustrative instances). Under UV irradiation of 30 mW·cm$^{-2}$, capsules containing a payload of 80 wt% P-HA in EPA solution showed a rapid photo-activation with 60% amine restoration in the first 10 min (Fig. 3a). The restoration rate plateaued after about 1 h photo irradiation and a maximum value of 95% was achieved. Given the efficient photoactivation, payload with 80 wt% protected-amines was selected for further studies to ensure an ample amine content, although the restoration efficiency further increased when the payload became less concentrated as dilution facilitated diffusion and light penetration (Supplementary Fig. 12a). When UV intensity was reduced to 15 mW·cm$^{-2}$, P-HA capsules presented almost an equally efficient amine restoration with a minimum deceleration of the photolysis kinetics. Nevertheless, a dramatic reduction of photo-activation efficiency was observed with a further decrease of UV intensity to 3 mW·cm$^{-2}$, a value corresponding to the sunlight UV strength (Air Mass 1.5 Spectra, ASTM G-173)[44]. The restoration rate remained below 35% with an extended UV exposure for 3 h (Supplementary Fig. 12b).

Photo-activation of P-EA (Fig. 3b) and P-PA (Fig. 3c) capsules was not as efficient due in part to their polarities and sterically hindered structures. Amine restoration rate of 78% and 62% was recorded for P-EA and P-PA capsules respectively with 30 mW·cm$^{-2}$ UV irradiation for 1 h. The activation of amines further slowed down over time because of the increased payload viscosity (Supplementary Table 1) and decreased light transmission through the capsules as the photolysis continued (Supplementary Fig. 10). All above factors made these two capsules more sensitive to the photo irradiation strength and reduced restoration rates of amines were observed with a 15 mW·cm$^{-2}$ UV light. Despite the various kinetics dictated by the types of amines, it is worthy mentioning that the spherical geometry as well as the high surface area of the microcapsules significantly enhanced photolysis by the remarkably enhanced payload exposure to UV light. In a control experiment, photo-activation of P-HA exhibited an exceedingly gradual pace when the same payload solution (i.e. 80 wt% P-HA in EPA) was directly exposed to the UV light within a quartz cuvette. At the 1 h checkpoint, less than 5% amine restoration was attained (Supplementary Fig. 13).

## Autonomous damage sensing based on photo-activated amine capsules

We then explored the potential of photo-activated amine capsules to facilitate the design of autonomous materials. In order to develop a polymer coating with the ability to self-report microscopic damage, we created microcapsules containing a 0.5 wt% solution of 4-chloro-7-nitrobenzofurazan (NBD-Cl) in EPA (see Methods for details). These capsules were combined with photo-activated HA (A-HA) capsules at concentrations 7.5 wt% and 2.5 wt%, respectively, and incorporated into a polydimethylsiloxane (PDMS) coating (Fig. 4a). Capsule concentration was selected based on our previous studies to ensure damage-sensing ability while minimizing impact on the polymer properties[20,39]. NBD-Cl serves as a reagent for fluorometric amine assays, resulting in the generation of a green fluorescence through a chemical reaction[45]. While a solution with simultaneously dissolved P-HA and NBD-Cl was non-emissive under illumination of UV light, a vibrant emission with a maximum at 530 nm was observed when photo activation was triggered to provide A-HA (Fig. 4b). The capsule-loaded PDMS specimen was scratched with a stylus and examined under white light and UV light using a stereomicroscopy (Fig. 4c). Damage-released A-HA excites the NBD-Cl indicator, generating a turn-on fluorescence that highlighted individual ruptured microcapsules. Intact areas outside the damage zone remained completely non-emissive, producing a significant contrast to amplify visual identification of the microcrack. This autonomous damage indication capability validated the chemical reactivity of the rejuvenated amines. A control PDMS coating loaded with P-HA capsules were evaluated in an identical fashion (Supplementary Fig. 14a) and the entire specimen was invisible under UV light with no fluorescence generated (Supplementary Fig. 14b). The dual-capsule enables damage reporting in polymer matrices that are not compatible with certain indicators, such as the insoluble NBD-Cl in PDMS.

Mechanically triggered release and subsequent reactivity of the photo-activated capsules were further examined in an epoxy coating that had been damaged through targeted ultrasound (Fig. 4d). A pH indicator, bromophenol blue (BPB), was dissolved in diglycidyl ether of bisphenol-A as a built-in colorimetric sensor of active amines. UV–vis absorption analysis revealed a color change of the BPB solution from yellow to purple when the added P-HA was activated via UV irradiation (Fig. 4e). Epoxy resin loaded with 2 wt% A-HA capsules, 0.15 wt% BPB, and 2 wt% triarylsulfonium hexafluorophosphate catalyst were casted on a glass substrate using doctor blade. A sonication probe was positioned on the surface of the epoxy film, emitting 20 kHz ultrasonic radiation at a power of 300 W for 2 seconds. This energy input induced internal damage and localized rupture of the capsules. Release of A-HA from the damaged capsules generated a purple color in the sonication region (Fig. 4f). The mechanochromic response highlights the distinctive release profile induced by the soundwave. No color change was observed in the ultrasound-impacted control specimen that contained P-HA capsules (Supplementary Fig. 15a), reassuring the photo-modulated mechanism of amine activation. Analogous coatings prepared with photo-activated EA and PA (noted as A-EA and A-PA) capsules were examined with the same sonication protocol, where the ultrasound-triggered damage was clearly discernible by the purple marks (Supplementary Fig. 15b, c). To highlight the adaptability of this technique, we also directed focused ultrasound towards an NBD-Cl solution containing A-HA capsules (Supplementary Fig. 16a) and administered a laser-induced shockwave to an epoxy film embedded with A-HA capsules and dispersed BPB molecules (Supplementary Fig. 16b). In both studies, the induced internal damage released active amines within the specimens, triggering visual indications that autonomously tracked the localized micro-defects.

## Recovery of adhesive strength at damaged interfaces

The application of amine capsules was also investigated by employing photo-activated polyfunctional A-EA as an epoxy crosslinker to develop adhesive strength at a damaged interface. As an initial experiment to demonstrate ability, a combination of A-HA capsules and epoxy capsules (Supplementary Fig. 17, EPON 813 epoxy resin as payload, see Methods for details) were mixed and crushed between

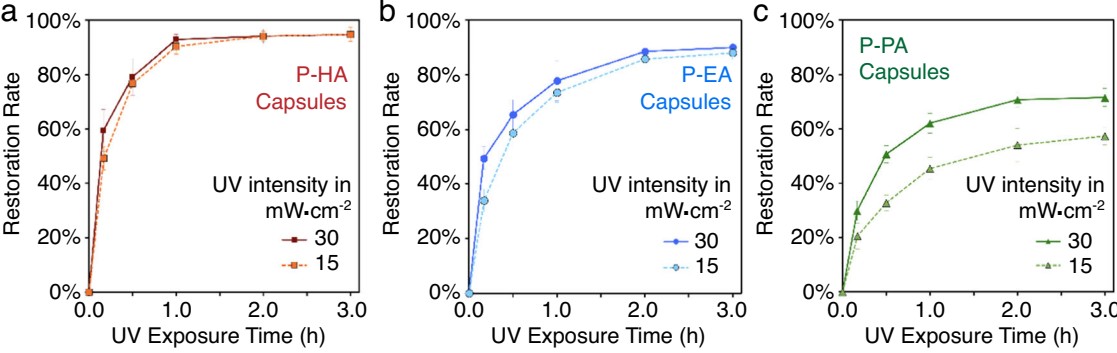

**Fig. 3 | Time-dependent amine restoration in capsules.** (**a**) NPPOC-protected hexylamine (P-HA), (**b**) NPPOC-protected 2,2'-(ethylenedioxy)bis(ethylamine) (P-EA), and (**c**) NPPOC-protected piperidine (P-PA) capsules under 365 nm UV light of different intensities. Data are presented as mean values ± SD ($n$ = 3 independent samples).

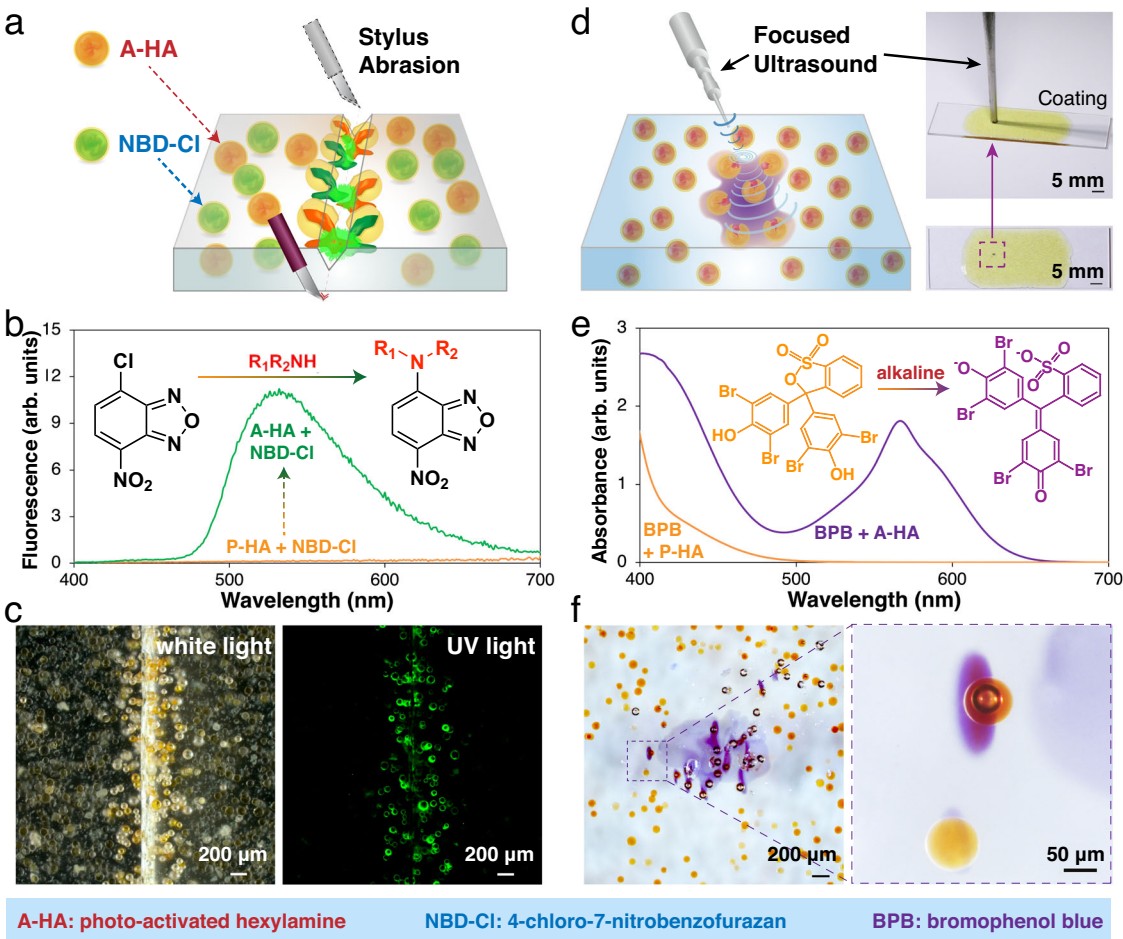

**Fig. 4 | Photo-activated amine capsules for autonomous damage indication in polymers. a** Schematic of a PDMS coating embedded with 7.5 wt% NBD-Cl capsules and 2.5 wt% A-HA capsules, exhibiting self-reporting ability to highlight microcracks. **b** Fluorescence emission spectra of NBD-Cl with P-HA and A-HA additions ($\lambda_{ex}$ = 365 nm). Inset illuminates the reaction mechanism between NBD-Cl and amines. **c** Stereomicrographs of scratched coating specimen under white light and UV light. Stereomicroscopy has a large depth of field, providing a comprehensive view of the microcapsules located at various depths within the specimen. **d** Schematic and experimental setup for study of ultrasound-induced internal damage in polymers. The specimen is prepared with dispersed A-HA capsules and built-in BPB molecules. **e** UV-Vis spectra of BPB with additions of P-HA and A-HA. Inset illuminates the sensing mechanism. **f** Stereomicrographs of the ultrasound-damaged spot in the epoxy matrix. The unique damage characteristic aroused by the soundwave is recorded.

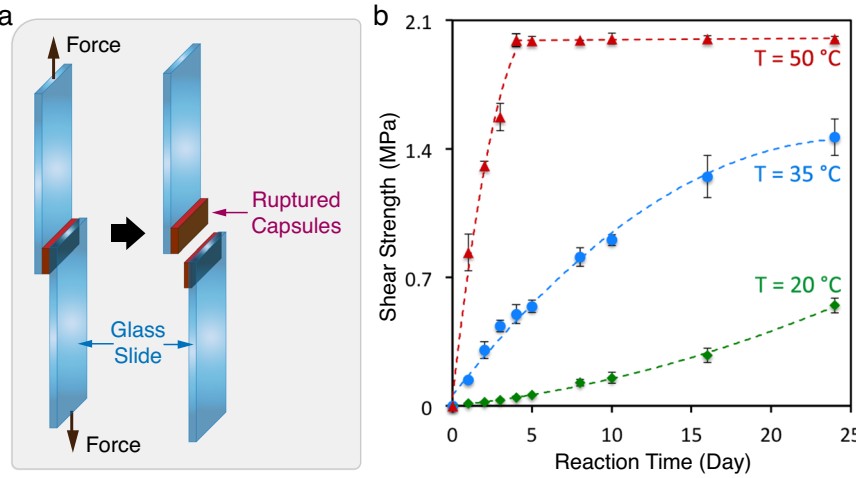

**Fig. 5 | Recovery of adhesive strength of a damaged interface. a** Schematic of lap shear test. A-EA capsules and epoxy capsules are ruptured between two glass slides to initiate adhesive strength. **b** Time-dependent shear strength development at various temperatures. Data are presented as mean values ± SD ($n = 3$ independent samples).

glass slides. The interfacial shear strength of the resulting bond formed between the glass slides was characterized by a lap shear experiment (Fig. 5a) with a range of curing protocols. While an observed room temperature (20 °C) curing demonstrated self-healing capability, the reaction showed slow kinetics due to the overall non-stoichiometric blending of the released payloads. The lack of mixing was significantly alleviated at elevated temperature, where a maximum shear strength of 2.0 MPa was achieved in 4 days at 50 °C (Fig. 5b), reaching 72% of the theoretical value (2.8 MPa) that was determined by a control experiment using pristine EA and epoxy resin (Supplementary Table 2). These results further illustrate the reactivity of photo-activated amines and highlight their versatility in developing a variety of adaptive materials.

## Discussion

We present a method for facile encapsulation of highly reactive organic bases and demonstrate its usefulness in creating smart polymers with autonomous functionalities. This scheme marries the concepts of on-demand photo-activation and in situ emulsification polymerization. We showed that chemical modification with a strategically selected, well-documented photo-removable protecting group effectively concealed the reactivity of aliphatic amines, producing responsive payloads that were compatible with the encapsulation solution. The prepared microcapsules exhibited excellent mechanical robustness, good thermal stability, and thin shell walls that ensured high amine content and facilitated light transmission. Amine restoration of intact capsules was achieved efficiently by UV irradiation. Both capsules before and after photoactivation presented exceptional stability, with no observed structural or performance degradation over the course of six months. Capsule storage in darkness is optimal to avoid any unexpected amine activation and maximize the shelf-life.

The reactivity of the photo-activated amine capsules was illustrated by initiating fluorogenic and chromogenic reactions in damaged polymers and restoring adhesive strength at damaged interfaces. Although pH indicators are appealing candidates for self-reporting polymers because of the colorimetric responses across the entire color spectrum, their practical implementation has been constrained due to technical challenges associated with microencapsulation. While there are a large number of available dyes, only a few have been successfully encapsulated[20,21], due to compatibility issues arising from their solubility in the payload and the pH value of the synthesis solution (Supplementary Fig. 18). The robust amine capsules coupled with the dispersed BPB in the epoxy matrix maximize the colorimetric damage-reporting by considerably reducing the background color

that would otherwise be present with BPB capsules (i.e. encapsulated BPB shows a vibrant green color because the pH of the encapsulation solution is the same as the color transition range (3.0 to 4.6) for BPB).

The versatility of this method was proved by successful encapsulation of a mono-functional primary amine, a polyfunctional primary amine, and a cyclic secondary amine. In theory, this strategy can be further extended to other amines, amino acids, nucleotides, and carbohydrates[36–38]. Our initial screening studies revealed that the photolysis kinetics are highly dependent on the characteristics of the payload, suggesting that molecular architecture, polarity, and viscosity are essential for specific applications, particularly those requiring highly concentrated active amines and flowable payloads. Meticulous administration of UV light is essential for effective amine activation. For P-HA and P-EA capsules reported in this study, exposure to a 365 nm UV light at an intensity of 30 mW·cm$^{-2}$ for 1 h ensures sufficient reactivity for damage-sensing and self-healing.

In contrast to the existing encapsulation techniques, this approach configured a photo trigger to regulate payload reactivities, which does not only extend longevity of the capsules but also sheds light on a smart core-shell structure with stimuli-responsive liquid cores. In addition to the photo-removable protecting groups, this concept integrates with alternative non-contact triggering mechanisms, allowing for payload activation through remote triggers that do not compromise the capsule integrity. To showcase this capacity, we masked a polyoxypropylene triamine (EPIKURE 3233) using a *tert*-butyloxycarbonyl (BOC) protecting group (Supplementary Fig. 19). Core-shell microcapsules containing the thermally responsive payload were prepared using the same in situ emulsification condensation polymerization method. Microwave was employed as an efficient activation approach to prevent overheating of the shell membranes, thus avoiding potential degradation of the capsules. The release of active amines from the mechanically ruptured capsules was verified through their colorimetric reaction with ninhydrin. We anticipate that readily available stimuli-removable protecting groups will make this method a useful tool for encapsulating a wide range of reactive chemicals such as bases, acids, and enzymes.

## Methods
### Materials
Hexylamine (HA, 99%)), 2,2′-(Ethylenedioxy)bis(ethylamine) (EA, 98%), piperidine (PA, ReagentPlus, 99%), 1-ethyl-2-nitrobenzene (96%), phosgene solution (15 wt% in toluene), potassium hydroxide (reagent grade, 90%), and paraformaldehyde (95%) from Sigma-Aldrich (St. Louis, MO) were used to prepare NPPOC-protected amines. Ethylene

maleic anhydride copolymer (EMA, Zemac-400, average molecular weight = 400,000) from Vertellus (Indianapolis, IN), and urea (ACS reagent, ≥99%), ammonium chloride (ACS reagent, ≥99.5%), resorcinol (ACS reagent, ≥99%), 1-octanol (ACS reagent, ≥99%), formaldehyde solution (ACS reagent, 37 wt% in $H_2O$), ethyl phenyl acetate (EPA, ReagentPlus, 99%)) from Sigma-Aldrich were used for synthesis of microcapsules. Epoxy resin EPON 813 from Miller-Stephenson (Houston, TX), 4-chloro-7-nitrobenzofurazan (NBD-Cl, HPLC, >98%) from TCI America (Portland, OR), and bromophenol blue (BPB, ACS reagent) from Sigma Aldrich were utilized to test the reactivity of the activated amines. Triarylsulfonium hexafluorophosphate (TSFP, 50% in propylene carbonate) obtained from Sigma Aldrich was used as a UV catalyst for curing epoxy resin EPON 828 (diglycidyl ether of bisphenol-A) in coating preparation. Silanol-terminated polydimethylsiloxane (DMS-S35) and poly(diethoxysiloxane) (PSI-021) from Gelest Inc. and tin catalyst (di-n-butyltin dilaurate (DBTL)) were used to prepare PDMS coatings. All chemicals were used as received with no further purification.

## Microencapsulation of protected aliphatic amines

Poly(urea-formaldehyde) (PUF) microcapsules were prepared via an emulsification polymerization method. 16 g protected aliphatic amines was mixed with 4 g EPA to form the core solution. 0.4 g 1,4-dimethoxybenzene (DMB) was added into the core solution as an internal reference to calculate UV-induced restoration rate of amines using $^1$H NMR. Under mechanical agitation at 800 rpm, the core solution was injected into an aqueous solution containing 125 mL 0.5 wt% EMA solution, 2.5 g urea, 0.25 g ammonium chloride, 0.25 g resorcinol, and 2 droplets of 1-octanol. After an emulsification of 10 min, 6.33 g formaldehyde solution was added into the emulsion and the encapsulation was carried out at 55 °C for 4 h with a heating rate of 1 °C/min. The prepared microcapsules were then filtered, rinsed, dried. Capsules with diameter ranging from 75 μm to 125 μm were isolated using a heavy-duty shaker that executed both horizontal circular and vertical tapping motions simultaneously for characterization and analysis.

## Photo-induced restoration of aliphatic amines in intact microcapsules

As-prepared microcapsules containing protected aliphatic amines were irradiated in an Intelliray UV flood system (Uvitron International, MA) for up to 3 h (Supplementary Fig. 20). Amine capsules were powder-like, which facilitated a mono-layer distribution on a reflective aluminum tray. UV intensity was precisely controlled by adjusting power of the UV lamp and sample distance from the light source. Reflective aluminum foil was employed on the side walls to ensure 360° UV irradiation. The UV intensities at various locations were verified using a UV light meter. A 365 nm UV light with intensities ranging from 3 mW cm$^{-2}$ to 30 mW cm$^{-2}$ were used in this study.

## Microencapsulation of NBD-Cl solution and epoxy resin

PUF microcapsules were prepared via a similar emulsification polymerization method. The encapsulation procedure is the same as described above while the payloads are 0.5 wt% NBD-Cl in EPA for damage indication capsules and EPON 813 for epoxy capsules.

## Characterization of amines and capsules

NMR spectra were recorded with a Varian INOVA 500 NB High-Resolution NMR spectrometer to examine the molecular structures of original, protected, and activated aliphatic amines. High-resolution mass spectrometry analyses were conducted using a Waters Synapt G2-Si ESI instrument. The size exclusion chromatographic assessments were carried out using an Agilent 1260 Infinity setup, which includes components such as an isocratic pump, a degassing unit, an automatic sampler, and a connected array of four Waters HR Styragel columns (dimensions 7.8 × 300 mm, models HR1, HR3, HR4, and HR5),

operating in tetrahydrofuran (THF) at a temperature of 25 °C and a consistent flow rate of 1 mL per minute. This setup incorporates a multifaceted detection mechanism, comprising an Agilent 1200 series G1362A Infinity Refractive Index Detector (RID), a Wyatt Viscostar II viscometric detector, and a Wyatt MiniDAWN Treos three-angle light scattering detector. The molecular weights and polydispersities (Đ) were evaluated using a 12-point standard calibration curve, utilizing narrowly distributed polystyrene standards with molecular weights ranging between 980 and 1,013,000 Dalton. The microstructure of the capsules was studied by SEM (FEI Quanta FEG 450 ESEM) and stereomicroscopy (Zeiss SteREO Discovery V20 Microscope). Thermal behaviors of core solution and microcapsules were investigated by thermogravimetric analysis (TA Instrument Q50) with 10 °C/min heating rate and nitrogen as purge gas. Fluorescence spectra (Horiba FluoroMax-4) of NBD-Cl were recorded from its THF solution with various additives including HA, P-HA, and A-HA. UV-Vis absorption spectra (Shimadzu UV-2401PC spectrometer) of BPB in THF solutions were measured with various additives including HA, P-HA, and A-HA.

## Self-reporting of mechanical damage in polymeric materials

PDMS coatings containing 2.5 wt% A-HA capsules and 7.5 wt% NBD-Cl capsules were casted on glass substrates using a doctor blade and cured overnight under ambient conditions. The coatings were damaged with a test panel scratcher (Corrocutter 639, Erichsen). Control coatings were prepared similarly with 2.5 wt% P-HA capsules and 7.5 wt% NBD-Cl capsules. To prepare epoxy coatings for autonomous indication of ultrasound-induced damage, 20 mg 15 min UV-irradiated A-HA (or A-EA, A-PA) capsules was dispersed in a precursor solution containing 0.85 g EPON 828, 0.15 g EPA, 0.02 g TSFP, and 1.5 mg BPB. The epoxy resin was casted on a glass slide with doctor blade and cured under 3 mW·cm$^{-2}$ 365 nm UV light for 40 min. The sonication probe was then placed on the epoxy surface. 20 kHz ultrasound with a power of 300 W was applied for 2 s. Control coatings were prepared in the same fashion with capsules containing protected amines.

## Characterization of adhesive strength by lap shear measurement

Activated A-EA capsules and epoxy capsules at 1-1 weight ratio were crushed between two glass slides with a 25 mm × 10 mm interfacial area. These glass slides were mounted in aluminum grips using an epoxy structural adhesive (Loctite M-31CL, McMaster-Carr), allowing pulling by a rail frame at a speed of 5 μm/s. Load vs. displacement relationship was collected after designed curing conditions and shear strength was calculated accordingly. Control specimens prepared by pristine EA and EPON 813 were tested. A solution consists of 1 g EPON 813, 0.21 g EA, and 0.79 g EPA was prepared and casted at the interfacial area between glass slides. The interfacial adhesion created by super glue (Loctite, Henkel Corporation) was also examined for comparison.

## Data availability

The data that support the findings of this study are available from the corresponding author upon request.

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

## Acknowledgements

The authors would like to acknowledge partial support for this work from the University of Illinois Center for Excellence in Self-Healing Materials,

funded by the Air Force Office of Scientific Research (FA9550-20-1-0194), and from China University of Petroleum (East China). We thank Dorothy Loudermilk for assistance with graphics and Dr. Ke Yang, Dr. Jaeuk Sung, Dr. Thu Doan, and Dr. Leon Dean for helpful discussions. Microscopy was performed using facilities maintained by the Imaging Technology Group at the Beckman Institute for Advanced Science and Technology at the University of Illinois Urbana-Champaign.

## Author contributions

W.L. conceived the study. X.L. designed the photo-induced deprotection scheme. X.L., C.S., and S.S. prepared the chemically protected amines. W.L., X.L., and B.J. studied the kinetics of amine restoration. W.L. synthesized the capsules, characterized the core-shell structures, designed and performed the damage sensing experiments. J.M.D. conducted the lap shear tests. N.R.S. and J.S.M. supervised the study. W.L. and N.R.S. wrote the manuscript with input from all co-authors.

## Competing interests

The authors declare no competing interests.
