## [Peer Review File · Nature Communications]

Photo-Modulated Activation of Organic Bases Enabling Microencapsulation and On-Demand ReactivityREVIEWER COMMENTS

Reviewer #1 (Remarks to the Author):

This manuscript by Li, Sottos, and co-workers presents a novel and conceptual method to encapsulate highly reactive chemicals. Amine microcapsules play essential roles in a wide range of applications including self-healing, damage-sensing, anti-corrosion, transient electronics, and multi-cycle carbon capture. Due to their vigorous reactivity, amines are challenging to encapsulate and prior efforts have achieved only partial success. This work reports on an innovative approach by placing a “photo switch” to regulate the reactivity of amines throughout microencapsulation. Through chemical modification using the photo-removable group NPOC, amines were rendered inert and compatible with the encapsulation system. The obtained amine capsules exhibited desirable properties and delivered on-demand reactivities with designed UV irradiation. This new method has demonstrated its versatility by examining three different types of amines and its effectiveness in enabling various damage detection and restoration schemes. The potential of using the vinyl compound from the photolysis to further enhance the self-healing capacity makes this work elegant with a well-thought-out design. This paper will be of great interest to a broad community and is well-suited for the broad readership of Nature Communications. Publication is recommended with a few suggestions to further elucidate the potential of this concept.

1. This work conceived a new scheme using removable protection groups to facilitate the microencapsulation of amines. The wide variety of available protection groups will likely make it useful in many applications beyond the scope of the current study. I can envision the development of additional remote activation schemes, such as those involving thermally removable protection groups. It is recommended that the authors provide their insight regarding alternative protection groups and the corresponding triggers in the Discussion section.
2. One merit of this method is its potential to encapsulate a wide range of amines. It will be helpful to discuss the potential of extending this approach to other polyfunctional amines.
3. One primary concern with amine capsules prepared by conventional methods is their stability. How does the photo-modulated activation scheme impact the stability of the capsules?
4. Did the kinetics of amine restoration inside the microcapsules slow down because of the shell membrane?
5. In the investigation of damage-sensing, the capsule amount appears to exceed the added value of 10 wt% (Figure 4c). Does this imply an uneven distribution of the microcapsules within the specimen?
6. How were the dyes selected for damage-sensing? There are other pH responsive dyes that have been used for damage detection (for example, 2',7'-dichlorofluorescein as shown in Adv. Mater. 2016, 28, 2189). Will the amine capsules work with such reported dyes? Is there anything special for BPB and NBD-Cl?
7. What was the rationale behind encapsulating NBD-Cl while dispersing BPB directly into the specimens?
8. What was the minimum crack size that the amine capsules were able to detect?
9. Microstructure of the epoxy capsules used for the adhesive strength development should be provided.
10. For the cited literature, the authors are recommended to learn more about the advantages and

disadvantages of the existing methods in the introduction session. Taking the microencapsulation method developed by Zhang et al. based on direct encapsulation of non-equilibrium droplets as an example, although "the collected microspheres were notably deformed", this does not necessarily mean "either a deficient liquid core or a poor structural stability". Authors should truthfully describe the advantages/disadvantages, or the existed problems that.

Reviewer #2 (Remarks to the Author):

Microcapsule-based damage detection or self-healing functionality in polymer materials is an attractive technology solving many molecular problems on the material level. Yet, many shortcomings impeded successful application of this principle up to today. In their manuscript, Sottos and coworkers introduce a new system for the microencapsulation of reactive bases in materials and demonstrate the functionality of the released functional groups for damage signalling as well as self-healing in polymers. Their approach comprises an encapsulating in situ emulsification polymerization method, which generates spherical nanometre sized and thermally stable capsules featuring thin and UV transparent walls that efficiently compartmentalize a solution of inactive light-activatable amines.

The manuscript is well-written, the structure is logical, and the work overall is presented in a way that is very much suitable for publication in Nature Communications. The employed chemistry is straightforward and draws from the vast pool of photoswitchable and photoactivatable bases available in the literature. The novelty of this system clearly lies in the manufacturing aspects rather than in the employed chemical methods. Very notable aspects of this work are the high thermal stability of the capsules and the long shelf life. Therefore, I believe this manuscript to be of transformative character warranting, in principle, its publication in Nature Communications.

However, certain parts of the manuscript should be revised that could otherwise be misleading or lack sufficient explanation and therefore require adjustments, providing of the according experimental data, or more concise language.

First, to clarify the difference in thermal stability between the core solution and the capsule in Figures 2 b and f, the x-axis could be adjusted (e.g., in an inset) to zoom in to the relevant temperature range around 300 °C, as the threshold temperatures for the core solutions are not given in the text.

Concerning irradiation experiments, an evolution of a brown color of the capsules is not a very robust visual indicator of photo activation as the desired irradiation reaction does not lead to the formation of compounds of brown color. The coloration results from unwanted degradation processes which would deserve a more thorough analysis and optimization of irradiation time and intensity and should be at least addressed as such in the text, especially since only a zoomed in part of the corresponding NMR spectrum is shown in the supporting information. This is also important with regards to the used internal proton NMR standard DMB and also EPA whose stability during 1 h of irradiation at 30 mW/cm² and 365 nm should be confirmed with the corresponding UV absorbance spectra and NMR analyses before and after irradiation of the pure compounds inside the capsules.

Additionally, the shelf-life of the system is estimated as "indefinite", however if even low UV light intensities suffice to activate the hexylamine species within 1 h, proper shielding from (sun)light is most

likely required for storage, which is especially important to note in the context of polymeric materials with regards to potential applications. Have the authors assessed a realistic application scenario with UV dosage?

Also a discussion of the increase in viscosity which differs for the respectively used amines would be desirable as this is only mentioned but not further discussed. Is there a molecular basis for this? Does the shell dissolve in the amine solution?

Concerning the mechanical behavior of the microcapsules, a more thorough analysis or discussion to elucidate the minimum force required to break the capsules is lacking. The capsules are only roughly described as “mechanically robust” but also capable of “early detection of mechanical damage” employing activation methods that seem like gentle application of mechanical stress, therefore, more detailed information by providing a mechanical characterization of the obtained material would be desirable. Maybe some nano- or microindentation or classical polymersome mechanical analysis, such as micropipette aspiration would work.

Moreover, the description within the text of Figure 4c stating that “every single ruptured microcapsule” is highlighted needs to be revised as this was not ensured. Instead, the sentence should be stating that single ruptured microcapsules are highlighted. Figure 4f is argued to reflect the “unique damage characteristic” of the applied ultrasound, this is also neither supported by any control experiment or simulation that compares to the resulting damage imaging, nor are the limitations addressed of stress mapping using this system, which are dictated by the size of the microcapsules as well as the flow of the released liquid.

In the conclusion, the photoprotecting group is referred to as photoswitch, which is not correct, as photoswitches can be reversibly interconverted. Additionally, the following phrase from the manuscript: “We showed that chemical modification with a photo-removable protecting group effectively concealed the reactivity of aliphatic amines” gives the impression as NPPOC as photoprotective group was newly introduced in this work, which is not the case.

Furthermore, in Supplementary Figure 4, the used concentration of the P-HA solution is missing and analytical data of the synthesized compounds generally lack the characterization by high-resolution mass spectrometry. Here, proton NMR spectra would also be desirable, specifically for the quantitative reaction of 2-(2-nitrophenyl)propan-1-ol with phosgene affording a brown oil, where again the brown color indicates impurities and thus most likely no full conversion of the compound.

Reviewer #3 (Remarks to the Author):

Revision manuscript NCOMMS-23-47591

Comments to the authors:

The authors present an innovative approach for efficiently encapsulating reactive organic amines and highlight its potential to confer polymers with functionalities. To achieve amine encapsulation, a photo-removable protective group is introduced in the amine compound, effectively masking their reactivity and rendering them compatible with the encapsulation process. Subsequently, the authors demonstrated the deprotection of the amines through UV irradiation, and examined the kinetics and

efficiency of this photodeprotection process. Finally, the authors demonstrated how these photoactivated reactive amines can be mechanically released to introduce new functionalities into polymers, including various reporting schemes as well as the formation/curing of adhesive between two substrates.

The concept of employing a stimuli-responsive core system to enable the encapsulation of components that would otherwise be too reactive to survive the process, is elegant and represent a novel framework within responsive materials based on encapsulation systems. Furthermore, this approach enhances control over the morphology of the capsules, imparting them with suitable mechanical properties and stability, aspects often challenging to achieve in other encapsulation techniques.

However, while the novelty of the approach can evidently be acknowledged, the presentation of the methodologies appears somewhat cursory and lacks depth, particularly with respect to the kinetics of the photoreactions. Mainly, I believe that the authors should incorporate additional controls to corroborate their statements. Furthermore, encapsulated systems have been widely employed to produce reporting materials, with more straightforward colorimetric concept than the approaches proposed here. In my opinion, the novelty of the framework could be elevated by demonstrating functionalities that were previously unattainable due to the challenges associated with encapsulating amines in mechanically robust capsules.

In summary, I recognize the novelty that the authors aim to highlight, but I believe the study and its demonstrations could be raised to meet the standards of Nature Communications. I would recommend this manuscript for publication, if the provided suggestions/concerns are properly addressed.

Please find below some suggestions to improve the manuscript:

- In the photochemistry section, the authors suggest that the photoactivation efficiency of P-EA and P-PA is hindered by their polarities and sterically hindered structures. Another consideration is the impact of viscosity on attenuating light transmission. While it is expected that all three factors would slow down the kinetics due to limited diffusion and light penetration, the temporal range of the kinetics provided in the manuscript does not allow for confirmation of these statements, leaving uncertainty regarding (for example) potential competitive side reactions or the influence of other factors, such as concentration and solvent. Upon closer examination of the kinetics, it is observed that the plateau is not achieved for P-EA and P-PA. Providing extended kinetics data and additional control experiments with concentration/solvent media changes may strengthen these arguments.

These additional insights would provide a better understanding of the thresholds required for specific functionalities, an aspect that seems somewhat lacking in this study. In brief, one could ask; what is the reaction performance needed to reveal the colorimetric output or to produce an efficient curing effect?

- Similar comments apply to the claim that the spherical geometry and high surface area enhance photolysis. The rationale behind the proposed controlled experiments is not entirely clear, particularly in terms of concentration considerations (bulk vs non-bulk). I suspect that reaction rates/efficiency may be influenced by concentration, which in turn affects viscosity and diffusion.

- The rationale behind the selection of capsules concentration for all demonstrators is not entirely clear to me. It would be beneficial to provide more insights into the motivation for these ratios.

- Furthermore, for the sake of reproducibility by peers, including an image of the photoprocess setup and providing additional details would be greatly helpful. The direction, distance, and intensity of light significantly influence the outcome of the photochemistry. I believe clarity in this regard is essential to avoid issues with reproducibility.

- The authors present various reporting systems that differ in their chemistry, presumably due to differences in the matrix. Perhaps the authors could offer some insights or comments on this matter.

Point-by-Point Response to Reviewer Comments

In the revised version of the manuscript entitled "Photo-Modulated Activation of Organic Bases Enabling Microencapsulation and On-Demand Reactivity", we addressed all the reviewers' questions and comments point-by-point. In the following section, we describe the specific changes/additions/corrections made to the manuscript based on reviewers' comments. Reviewer comments are listed in blue, while our responses follow in plain text, and changes/additions to the manuscript text are shown in "quoted italics".

Response to Reviewer 1

This manuscript by Li, Sottos, and co-workers presents a novel and conceptual method to encapsulate highly reactive chemicals. Amine microcapsules play essential roles in a wide range of applications including self-healing, damage-sensing, anti-corrosion, transient electronics, and multi-cycle carbon capture. Due to their vigorous reactivity, amines are challenging to encapsulate and prior efforts have achieved only partial success. This work reports on an innovative approach by placing a "photo switch" to regulate the reactivity of amines throughout microencapsulation. Through chemical modification using the photo-removable group NPPOC, amines were rendered inert and compatible with the encapsulation system. The obtained amine capsules exhibited desirable properties and delivered on-demand reactivities with designed UV irradiation. This new method has demonstrated its versatility by examining three different types of amines and its effectiveness in enabling various damage detection and restoration schemes. The potential of using the vinyl compound from the photolysis to further enhance the self-healing capacity makes this work elegant with a well-thought-out design. This paper will be of great interest to a broad community and is well-suited for the broad readership of Nature Communications. Publication is recommended with a few suggestions to further elucidate the potential of this concept.

Response: We thank the reviewer for the positive evaluation of our work.

1. This work conceived a new scheme using removable protection groups to facilitate the microencapsulation of amines. The wide variety of available protection groups will likely make it useful in many applications beyond the scope of the current study. I can envision the development of additional remote activation schemes, such as those involving thermally removable protection groups. It is recommended that the authors provide their insight regarding alternative protection groups and the corresponding triggers in the Discussion section.

Response: This concept can indeed be integrated with protecting groups that respond to other stimuli. To ensure capsule integrity, non-contact triggers capable of remotely activating the payload while maintaining the shell stability are ideal. In contrast to photoactivation schemes, the employment of thermally removable protecting groups requires precise control over the triggering temperature threshold, as excessive heat may degrade the shell membranes. In addition to conventional heating methods, microwave and magnetic hyperthermia are potential alternatives due to their high efficiency and targeted heating ability. To demonstrate the capability, we have encapsulated a polyoxypropylene triamine using a *tert*-butyloxycarbonyl (BOC) protecting group to regulate the reactivity of the payloads. We have included the following discussion and Supplementary Fig. 18 in the revised manuscript on pages 17-18.

"In addition to the photo-removable protecting groups, this concept integrates with alternative non-contact triggering mechanisms, allowing for payload activation through remote triggers that do not compromise the capsule integrity. To showcase this capacity, we masked a polyoxypropylene triamine (EPIKURE 3233) using a tert-butyloxycarbonyl (BOC) protecting group (Supplementary Fig. 18). Core-shell microcapsules containing the thermally responsive payload were prepared using the same in situ emulsification condensation polymerization method. Microwave was employed as an efficient activation approach to prevent overheating of the shell membranes, thus avoiding potential degradation of the capsules. The release of active amines from the mechanically ruptured capsules was verified through their colorimetric reaction with ninhydrin."

2. One merit of this method is its potential to encapsulate a wide range of amines. It will be helpful to discuss the potential of extending this approach to other polyfunctional amines.

Response: Yes, the protecting group NPPOC has been utilized to mask reactivities of a broad range of amines, amino acids, nucleotides, and carbohydrates (*Chem. Rev.* 2013, 113, 1, 119–191, doi.org/10.1021/cr300177k; *Chem. Rev.* 2009, 109, 6, 2455–2504, doi.org/10.1021/cr800323s), all of which can theoretically be integrated with the *in situ* microencapsulation technique. In our early screening studies, polyfunctional amines such as 1,3-diaminopropane and diethylenetriamine were encapsulated. These amines showed increased viscosity and reduced activation efficiency upon NPPOC protection, posing notable challenges for microencapsulation and subsequent delivery. We noticed that the photolysis kinetics are largely dependent on the molecular characteristics and payload attributes. These factors shall be carefully examined for specific applications, especially those requiring highly concentrated active amines and flowable payloads, such as in damage-sensing and self-healing.

We have added the following sentences in the Discussion section (page 17) to highlight the potential of this method.

"In theory, this strategy can be further extended to other amines, amino acids, nucleotides, and carbohydrates.³⁶⁻³⁸ Our initial screening studies revealed that the photolysis kinetics are highly dependent on the characteristics of the payload, suggesting that molecular architecture, polarity, and viscosity are essential for specific applications, particularly those requiring highly concentrated active amines and flowable payloads."

3. One primary concern with amine capsules prepared by conventional methods is their stability. How does the photo-modulated activation scheme impact the stability of the capsules?

Response: One notable advantage of this encapsulation approach is the exceptional stability of the amine capsules. All capsules investigated in this study were isolated from the synthesis solution, thoroughly rinsed to remove surface impurities, dried under ambient conditions, and subjected to vigorous sieving using a heavy-duty shaker that executed both horizontal circular and vertical tapping motions simultaneously. As indicated by the optical and SEM micrographs, all capsules retained excellent structural integrity throughout these processing steps. Moreover, the microstructural and thermal analyses revealed that these capsules survived strong UV irradiations. During coating preparation, additional processing steps such as mechanical blending, casting, and curing were carried out, none of which compromised the capsules. The amine capsules and

damage-sensing specimens, stored for over 6 months, have exhibited sustained structural integrity and performance comparable to their initial state.

We have added descriptions in the Methods section (page 19) to provide additional processing details and highlighted the improved stability in the Discussion section (page 16).

"The prepared microcapsules were then filtered, rinsed, and dried. Capsules with diameter ranging from 75 μm to 125 μm were isolated using a heavy-duty shaker that executed both horizontal circular and vertical tapping motions simultaneously for characterization and analysis. "

"Both capsules before and after photoactivation presented exceptional stability, with no observed structural or performance degradation over the course of six months."

4. Did the kinetics of amine restoration inside the microcapsules slow down because of the shell membrane?

Response: The shell membrane, with a thickness of approximately 100 nm, allows for sufficient UV penetration. In fact, a side-by-side comparison between the photolysis kinetics of the same P-HA solution in capsules versus in quartz cuvette was conducted, revealing that the small-scale core-shell structure actually facilitates the photoactivation. The power of light received per unit volume increases with the specific surface area, and the capsules are subjected to approximately 2 orders of magnitude stronger UV irradiation than the solution in the cuvette (i.e. 9677 mW/mL for capsules vs. 90 mW/mL for cuvette solution with UV intensity at 15 mW $\cdot\text{cm}^{-2}$).

The experimental results along with the theoretical analysis are provided in Supplementary Fig. 12.

5. In the investigation of damage-sensing, the capsule amount appears to exceed the added value of 10 wt% (Figure 4c). Does this imply an uneven distribution of the microcapsules within the specimen?

Response: The total capsule loading is indeed 10 wt% (7.5 wt% NBD-Cl capsule and 2.5 wt% A-HA capsule) in Fig. 4c. Capsules were well-dispersed in the polymer by controlling mixing and viscosity to ensure a homogeneous distribution. The image was captured by a stereomicroscope that offered extended depth of field, allowing for a relatively extensive range to be simultaneously in focus. This means that the image in Fig. 4c shows microcapsules located at various depths within the specimen rather than being confined to a single plane or cross-section.

We have added a description in the caption of Fig. 4 to clarify it.

"Stereomicroscopy has a large depth of field, providing a comprehensive view of the microcapsules located at various depths within the specimen."

6. How were the dyes selected for damage-sensing? There are other pH responsive dyes that have been used for damage detection (for example, 2',7'-dichlorofluorescein as shown in Adv. Mater. 2016, 28, 2189). Will the amine capsules work with such reported dyes? Is there anything special for BPB and NBD-Cl?

Response: We thank the reviewer for this comment. The highly stable amine capsules obtained in this study can be used with other dyes such as 2',7'-dichlorofluorescein. In this work, we selected a different pH-sensitive dye, BPB, because of the challenges using this dye in our prior work. Encapsulation of BPB results in capsules that exhibit a strong green color due to the pH of the encapsulation solution which is the same as the color transition range of BPB (3.0-4.6). The green color of the microcapsules compromises the colorimetric contrast and limits applications where aesthetics is important, such as automotive coatings. The amine capsules enable the use of BPB as a strong colorimetric sensor in polymeric materials and mitigate the background color issue. To further demonstrate the versatility of these capsules, we utilized NBD-Cl as a model compound for developing fluorogenic damage-sensing polymers. Other reagents for fluorometric amine assays, such as fluorescamine, also work well with these amine capsules.

We have clarified the Discussion section on pages 16-17 and control experiments (Supplementary Fig. 17) to better describe the rationale of dye selection.

7. What was the rationale behind encapsulating NBD-Cl while dispersing BPB directly into the specimens?

Response: As stated in our response to point #6, encapsulation of BPB results in capsules that exhibit a strong green color. Hence, we chose to disperse BPB in the epoxy matrix to maximize the colorimetric damage-reporting, reducing the background color that would otherwise be present with BPB capsules. In contrast, the insolubility of NBD-Cl in the PDMS matrix necessitated its encapsulation.

We have added descriptions regarding BPB in the Discussion section (pages 16-17) and NBD-Cl in "damage-sensing" of the Results section on page 12.

"The robust amine capsules coupled with the dispersed BPB in the epoxy matrix maximize the colorimetric damage-reporting by considerably reducing the background color that would otherwise be present with BPB capsules (i.e. encapsulated BPB shows a vibrant green color because the pH of the encapsulation solution is the same as the color transition range (3.0 to 4.6) for BPB)."

"The dual-capsule enables damage reporting in polymer matrices that are not compatible with certain indicators, such as the insoluble NBD-Cl in PDMS."

8. What was the minimum crack size that the amine capsules were able to detect?

Response: The limits of microcrack detection were presented in our previous studies (ACS Cent. Sci. 2016, 2, 598; Adv. Mater. 2016, 28, 2189; ACS Appl. Mater. Interfaces 2018, 10, 40361). We've shown that microcracks as small as 2 microns in width can be detected.

9. Microstructure of the epoxy capsules used for the adhesive strength development should be provided.

Response: We thank the reviewer for pointing out this omission. An optical micrograph and an SEM image of the epoxy capsules have been added as the Supplementary Fig. 16.

10. For the cited literature, the authors are recommended to learn more about the advantages and disadvantages of the existing methods in the introduction session. Taking the microencapsulation method developed by Zhang et al. based on direct encapsulation of non-equilibrium droplets as an example, although "the collected microspheres were notably deformed", this does not necessarily mean "either a deficient liquid core or a poor structural stability". Authors should truthfully describe the advantages/disadvantages, or the existed problems that.

Response: We have revised the Introduction section on page 4 and included both advantages and disadvantages of the technique.

"Zhang et al. coupled the above synthesis solution with a microfluidic device.²⁸ Individual micro-droplets were deposited in the reaction solution, leading to a rapid encapsulation via interfacial polymerization before an equilibrium emulsion was formed. Capsules with a polyamine payload were successfully prepared, but the polyurea shell wall was deformed due to shrinkage during the rapid shell formation process."

Response to Reviewer 2

Microcapsule-based damage detection or self-healing functionality in polymer materials is an attractive technology solving many molecular problems on the material level. Yet, many shortcomings impeded successful application of this principle up to today. In their manuscript, Sottos and coworkers introduce a new system for the microencapsulation of reactive bases in materials and demonstrate the functionality of the released functional groups for damage signalling as well as self-healing in polymers. Their approach comprises an encapsulating in situ emulsification polymerization method, which generates spherical nanometre sized and thermally stable capsules featuring thin and UV transparent walls that efficiently compartmentalize a solution of inactive light-activatable amines.

The manuscript is well-written, the structure is logical, and the work overall is presented in a way that is very much suitable for publication in Nature Communications. The employed chemistry is straightforward and draws from the vast pool of photoswitchable and photoactivatable bases available in the literature. The novelty of this system clearly lies in the manufacturing aspects rather than in the employed chemical methods. Very notable aspects of this work are the high thermal stability of the capsules and the long shelf life. Therefore, I believe this manuscript to be of transformative character warranting, in principle, its publication in Nature Communications.

However, certain parts of the manuscript should be revised that could otherwise be misleading or lack sufficient explanation and therefore require adjustments, providing of the according experimental data, or more concise language.

Response: We appreciate the reviewer's positive comments. We've revised the manuscript with additional description/discussion and experimental results.

First, to clarify the difference in thermal stability between the core solution and the capsule in Figures 2 b and f, the x-axis could be adjusted (e.g., in an inset) to zoom in to the relevant temperature range around 300 °C, as the threshold temperatures for the core solutions are not given in the text.

Response: We have added insets in Fig. 2b and 2f to provide magnified views of thermogravimetric analyses around 300 °C. The caption has been updated accordingly.

The onset weight loss in TGA depends on various factors, including boiling point and evaporation rate of the core solution, gas flow rate, and thermal stability of the shell materials. The weight losses of the core solutions were delayed when heated within the sealed microcontainers (i.e. microcapsules) until the temperature reached the critical value (180 °C) that compromised the integrity of the core-shell structure. These results confirm the successful microencapsulation and demonstrate the excellent thermal stability of the microcapsules.

Concerning irradiation experiments, an evolution of a brown color of the capsules is not a very robust visual indicator of photo activation as the desired irradiation reaction does not lead to the formation of compounds of brown color. The coloration results from unwanted degradation processes which would deserve a more thorough analysis and optimization of irradiation time and intensity and should be at least addressed as such in the text, especially since only a zoomed in part of the corresponding NMR spectrum is shown in the supporting information. This is also

important with regards to the used internal proton NMR standard DMB and also EPA whose stability during 1 h of irradiation at 30 mW/cm² and 365 nm should be confirmed with the corresponding UV absorbance spectra and NMR analyses before and after irradiation of the pure compounds inside the capsules.

Response: The photolysis mechanism of NPPOC-protected amines is well documented in the literature (Macromolecules 2014, 47, 18, 6159–6165, doi.org/10.1021/ma501366f; Chem. Rev. 2013, 113, 1, 119–191; doi.org/10.1021/cr300177k; J. Am. Chem. Soc. 2009, 131, 4, 1513–1522, doi.org/10.1021/ja807612y) and is depicted as follows:

The complex mechanism involves multiple excited states and free radicals. Formation of the primary byproduct, 2-(2'-nitrophenyl)propene (compound 5), was confirmed in our NMR studies of the neat P-HA (Supplementary Fig. 1) and P-HA capsules (Supplementary Fig. 2), with no other detectable compounds identified. This byproduct is a highly reactive nitro compound with an extended conjugation length compared to the structure of the protected amine. The strong electron-withdrawing effect of the nitro group and the delocalization of the double bond intrinsically red-shift the absorption of compound 5, resulting in visible coloration. Compound 5 has been reported in the literature as a yellow (Angew. Chem. Int. Ed., 54: 11809, doi.org/10.1002/anie.201505713) or brown oil (ACS Catal. 2017, 7, 8, 5518, doi.org/10.1021/acscatal.7b01915) with clean NMR spectra, prepared via different synthetic routes. Additionally, derivatives of compound 5, such as 2-nitrostyrene (Angew. Chem. Int. Ed. 2018, 57, 16431, doi.org/10.1002/anie.201810328), 3-nitrostyrene (Sigma Aldrich, N26601), and 4-nitrostyrene (TCI America, N0503), are reported or sold as yellow oils. The inherent color of this byproduct is one reason for the brown color observed in the photoactivated payloads. Another potential reason is the incomplete deprotection of NPPOC. Most studies related to NPPOC have focused on dilute solutions or surface applications, with limited research on highly concentrated state. In our microcapsules, the payloads contain 80 wt% P-HA, and such a high concentration under intense UV irradiation might delay and impede internal hydrogen migration (from intermediates 1 to 2, 3 to 4) and the intersystem conversion (from intermediates 2 to 3). This could lead to the formation of intermediates with radicals and enhanced conjugations, possibly resulting in intense colors.

We have examined NMR and UV-Vis spectra of EPA, DMB, and a 2 wt% DMB in EPA solution, confirming the stability of the solvent and the reference throughout the exposure to a 365 nm light at an intensity of 30 mW·cm⁻² for 1 h (Supplementary Fig. 4,5,6). Control capsules using pure EPA

and the 2 wt% DMB in EPA as the core solutions were also prepared. NMR analyses before and after the same UV irradiation further demonstrated the stability of EPA and DMB within the microcapsules (Supplementary Fig. 7).

We have provided NMR spectra, UV-Vis spectra, photoactivation mechanism, and additional control experiments in Supplementary Fig. 1-7, and revised the descriptions on page 9 of the manuscript based on these studies.

"The capsules acquired a brown color upon exposure to the UV light (Fig. 2h), which was attributed to the photolysis byproduct 2-(2'-nitrophenyl)propene and the potential intermediates (Supplementary Fig. 1).⁴²⁻⁴³ Payload solutions were extracted from the UV-irradiated P-HA capsules and analyzed with ¹H NMR (Supplementary Fig. 2), providing evidence of the photo-induced removal of the NPPOC protecting group and confirming the well-documented photolysis mechanism (Supplementary Fig. 3). Additional chemical and optical analyses of the solvent EPA, the internal NMR reference DMB, and control capsules containing such compounds confirmed the stability of these additives throughout the UV exposure (Supplementary Fig. 4-7)."

Additionally, the shelf-life of the system is estimated as "indefinite", however if even low UV light intensities suffice to activate the hexylamine species within 1 h, proper shielding from (sun)light is most likely required for storage, which is especially important to note in the context of polymeric materials with regards to potential applications. Have the authors assessed a realistic application scenario with UV dosage?

Response: We have revised the sentence by saying this method "significantly" extended the shelf-life of microcapsules. A comment has also been added to the Discussion section on page 16, indicating that the shelf-life could be considerably increased by storing these capsules in darkness.

"Capsule storage in darkness is optimal to avoid any unexpected amine activation and maximize the shelf-life."

To better describe capsule behaviors under ambient conditions, we examined the photoactivation of P-HA capsules using a 365 nm UV lamp with an intensity of 3 mW·cm⁻², a value corresponding to the UV strength of sunlight (sum of wavelengths from 280 nm to 400 nm) as per the Air Mass 1.5 Spectra (ASTM G-173). Upon exposure to this mild UV light, a dramatic reduction in photoactivation efficiency was observed in comparison to the photolysis kinetics under intense UV irradiation at 15 mW·cm⁻² and 30 mW·cm⁻² (Supplementary Fig. 11b). In a practical application scenario, where protected amine capsules are embedded in polymer matrices, the UV exposure would be even further diminished compared to sunlight. As a result, we've not experienced unwanted photoactivation during the routine handling of these capsules.

Also a discussion of the increase in viscosity which differs for the respectively used amines would be desirable as this is only mentioned but not further discussed. Is there a molecular basis for this? Does the shell dissolve in the amine solution?

Response: This is a good point by the reviewer. We have conducted a series of preliminary experiments to explore the cause of the viscosity increase. The neat protected-amines exhibited an increase in viscosity upon UV irradiation, ruling out the solvent EPA and the NMR reference DMB as potential causes. Furthermore, NMR spectra of the photoactivated amines did not show any

noticeable peak broadening or peaks indicative of polymer/oligomer backbones (Supplementary Fig. 1 and Fig. 2), suggesting the byproduct 2-(2'-nitrophenyl)propene did not undergo polymerization.

The increase in viscosity could be the result of association between charged species, such as carbamate anion intermediate and the ammonium cation of the amine (J. Am. Chem. Soc. 1980, 102, 9, 3072, doi.org/10.1021/ja00529a033). As shown, the photolysis mechanism of NPPOC-protected amines involves multiple proton transfer steps. Under intense UV irradiation, the proton transfer in a highly concentrated environment may not proceed as efficiently as in a dilute solution. The carbamate intermediate 6 may associate with ammonium cation of the amine to form a charged complex. This type of complex has been documented in prior studies, specifically in the context of the reaction between amines and carbon dioxide (J. Am. Chem. Soc. 2012, 134, 33, 13834, doi.org/10.1021/ja304888a; ACS Omega 2020, 5, 40, 26125, doi.org/10.1021/acsomega.0c03727).

Due to the high degree of crosslinking in the capsule shell, dissolution in the payloads is unlikely. However, the shell materials, predominantly poly(urea-formaldehyde), can establish hydrogen bonds with polar compounds in the payloads, which could further increase the viscosity. The molecular structure and polarity of specific amines contribute to the involvement in these potential mechanisms.

Given that (1) the examination of the hypotheses requires specialized experiments, as characterization of diluted payloads or unencapsulated solutions will inevitably introduce additional variables, and (2) the viscosity changes do not impact the key conclusions of this study or capsule performance in damage-sensing and interface healing, we believe the mechanism of the viscosity is beyond the scope of this paper and is the subject of future investigation.

We have included the photoactivation mechanism as Supplementary Fig. 3 to illustrate possible intermediates and charged species.

Concerning the mechanical behavior of the microcapsules, a more thorough analysis or discussion to elucidate the minimum force required to break the capsules is lacking. The capsules are only roughly described as “mechanically robust” but also capable of “early detection of mechanical damage” employing activation methods that seem like gentle application of mechanical stress, therefore, more detailed information by providing a mechanical characterization of the obtained material would be desirable. Maybe some nano- or microindentation or classical polymersome mechanical analysis, such as micropipette aspiration would work.

Response: The mechanical properties of the poly(urea-formaldehyde) microcapsules have been examined in our earlier studies (Exp. Mech. 2006, 46, 725). Young's modulus of the capsule shell wall was determined to be 3.7 GPa. Microcapsule diameter had significant effect on failure strength, with values increasing from 0.24 MPa to 0.80 MPa as the diameter decreased from 187 μm to 65 μm .

In this study, the term "mechanical robustness" is used to emphasize that the capsules survive the manufacturing process. In the preparation of the damage-sensing specimens, the microcapsules were isolated from the synthesis solution, thoroughly rinsed to remove surface impurities, dried under ambient conditions, subjected to vigorous sieving using a heavy-duty shaker that executed both horizontal circular and vertical tapping motions simultaneously, and then blended with polymer resins followed by curing. The microstructural and thermal analyses confirmed that these capsules were robust enough to withstand these processes.

The term "early detection of mechanical damage" in this context refers to the capability to visualize damage at small-scales, and it doesn't necessarily imply fragility in the capsules. It highlights the sensitivity of the system in detecting even microscale mechanical damage. When polymers undergo mechanical stress, microcracks propagate through the matrices, rupture the embedded capsules, and release the payloads, triggering a colorimetric reporting response.

We have added the following sentence on page 7 to better describe the mechanical property of the capsules.

"The poly(urea-formaldehyde) shell wall has a Young's modulus of ~ 3.7 GPa,⁴¹ and was robust enough to survive processing conditions."

Moreover, the description within the text of Figure 4c stating that "every single ruptured microcapsule" is highlighted needs to be revised as this was not ensured. Instead, the sentence should be stating that single ruptured microcapsules are highlighted. Figure 4f is argued to reflect the "unique damage characteristic" of the applied ultrasound, this is also neither supported by any control experiment or simulation that compares to the resulting damage imaging, nor are the limitations addressed of stress mapping using this system, which are dictated by the size of the microcapsules as well as the flow of the released liquid.

Response: We thank the reviewer for raising this point and we revised the manuscript on page 12 and page 14 as follows:

"Damage-released A-HA excites the NBD-Cl indicator, generating a turn-on fluorescence that highlighted individual ruptured microcapsules."

"The mechanochromic response highlights the distinctive release profile induced by the soundwave."

In the conclusion, the photoprotecting group is referred to as photoswitch, which is not correct, as photoswitches can be reversibly interconverted. Additionally, the following phrase from the manuscript: "We showed that chemical modification with a photo-removable protecting group effectively concealed the reactivity of aliphatic amines" gives the impression as NPPOC as photoprotective group was newly introduced in this work, which is not the case.

Response: We have replaced the term "switch" with "trigger", and revised the other sentence to explicitly state that the NPPOC protecting group was chosen from the existing pool on page 16.

"this approach configured a photo trigger to regulate payload reactivities"

"We showed that chemical modification with a strategically selected, well-documented photo-removable protecting group effectively concealed the reactivity of aliphatic amines."

Furthermore, in Supplementary Figure 4, the used concentration of the P-HA solution is missing and analytical data of the synthesized compounds generally lack the characterization by high-resolution mass spectrometry. Here, proton NMR spectra would also be desirable, specifically for the quantitative reaction of 2-(2-nitrophenyl)propan-1-ol with phosgene affording a brown oil, where again the brown color indicates impurities and thus most likely no full conversion of the compound.

Response: Concentration of the P-HA in EPA solution is 80 wt%, which has been included in the caption of Supplementary Fig. 12. This control experiment maintained a constant concentration for both samples, allowing for a direct comparison of the geometry effect. Due to the high specific surface area, the solution experienced an approximate two orders of magnitude increase in light power per unit volume when stored within the small-scale core-shell structures compared to a cuvette.

High-resolution mass spectrometry (HRMS) analyses of the final products P-HA, P-EA, and P-PA have now been incorporated into the synthetic procedures in the Supplementary Information. Compound 3, NPPOC-Cl, was synthesized using phosgene gas, a reagent known for its high toxicity. Owing to safety concerns, the crude product was used directly in the subsequent step without further purification, isolation, or characterization, thereby minimizing potential exposure to hazards. The resulting brown oil likely contained impurities; however, these impurities did not adversely affect the subsequent reactions, which proceeded successfully with decent yields. The phrase "(quantitative yield)" has been omitted to prevent confusion.

Response to Reviewer 3

The authors present an innovative approach for efficiently encapsulating reactive organic amines and highlight its potential to confer polymers with functionalities. To achieve amine encapsulation, a photo-removable protective group is introduced in the amine compound, effectively masking their reactivity and rendering them compatible with the encapsulation process. Subsequently, the authors demonstrated the deprotection of the amines through UV irradiation, and examined the kinetics and efficiency of this photodeprotection process. Finally, the authors demonstrated how these photoactivated reactive amines can be mechanically released to introduce new functionalities into polymers, including various reporting schemes as well as the formation/curing of adhesive between two substrates.

The concept of employing a stimuli-responsive core system to enable the encapsulation of components that would otherwise be too reactive to survive the process, is elegant and represent a novel framework within responsive materials based on encapsulation systems. Furthermore, this approach enhances control over the morphology of the capsules, imparting them with suitable mechanical properties and stability, aspects often challenging to achieve in other encapsulation techniques.

Response: We thank the reviewer for the positive evaluation of our work.

However, while the novelty of the approach can evidently be acknowledged, the presentation of the methodologies appears somewhat cursory and lacks depth, particularly with respect to the kinetics of the photoreactions. Mainly, I believe that the authors should incorporate additional controls to corroborate their statements. Furthermore, encapsulated systems have been widely employed to produce reporting materials, with more straightforward colorimetric concept than the approaches proposed here. In my opinion, the novelty of the framework could be elevated by demonstrating functionalities that were previously unattainable due to the challenges associated with encapsulating amines in mechanically robust capsules.

In summary, I recognize the novelty that the authors aim to highlight, but I believe the study and its demonstrations could be raised to meet the standards of Nature Communications. I would recommend this manuscript for publication, if the provided suggestions/concerns are properly addressed.

Response: We have included additional control experiments to further understand the kinetics, which are elaborated in the following point-by-point response to specific comments.

The reviewer made a great point regarding the impact of this encapsulation strategy on colorimetric damage reporting. We acknowledge that various self-reporting systems are available. In fact, our group has developed a few of them, including the widely used 2',7'-dichlorofluorescein (DCF). The development of mechanically robust and thermally stable amine capsules will significantly expand the colorimetric reporting library by easily pairing these amine capsules with various pH-sensitive dyes dispersed in polymer matrices. The obstacle in establishing such a simple yet effective mechanism was the potential for false positives, stemming from the instability of previously reported amine capsules. Encapsulation of pH-sensitive dyes is limited to specific candidates due to the compatibility issues arising from their solubility in the payload and the pH value of the synthesis solution. In the revised manuscript, we've incorporated control experiments and

expanded our discussion to highlight the capability of the newly synthesized amine capsules in creating self-reporting polymers. Capsules containing BPB exhibited a strong green color because pH of the synthesis solution falls within the color transition range, which precluded applications demanding high colorimetric contrast and prioritizing aesthetics (e.g. automotive coatings). We show that the employment of the robust amine capsules coupled with BPB as the built-in sensing molecules significantly enhanced the colorimetric response by minimizing the background color. The following sentences have been added in the Discussion section (pages 16-17) and additional control experiments have been included as Supplementary Fig. 17.

"Although pH indicators are appealing candidates for self-reporting polymers because of the colorimetric responses across the entire color spectrum, their practical implementation has been constrained due to technical challenges associated with microencapsulation. While there are a large number of available dyes, only a few have been successfully encapsulated,^{20,21} due to compatibility issues arising from their solubility in the payload and the pH value of the synthesis solution (Supplementary Fig. 17). The robust amine capsules coupled with the dispersed BPB in the epoxy matrix maximize the colorimetric damage-reporting by considerably reducing the background color that would otherwise be present with BPB capsules (i.e. encapsulated BPB shows a vibrant green color because the pH of the encapsulation solution is the same as the color transition range (3.0 to 4.6) for BPB)."

Please find below some suggestions to improve the manuscript:

- In the photochemistry section, the authors suggest that the photoactivation efficiency of P-EA and P-PA is hindered by their polarities and sterically hindered structures. Another consideration is the impact of viscosity on attenuating light transmission. While it is expected that all three factors would slow down the kinetics due to limited diffusion and light penetration, the temporal range of the kinetics provided in the manuscript does not allow for confirmation of these statements, leaving uncertainty regarding (for example) potential competitive side reactions or the influence of other factors, such as concentration and solvent. Upon closer examination of the kinetics, it is observed that the plateau is not achieved for P-EA and P-PA. Providing extended kinetics data and additional control experiments with concentration/solvent media changes may strengthen these arguments.

These additional insights would provide a better understanding of the thresholds required for specific functionalities, an aspect that seems somewhat lacking in this study. In brief, one could ask; what is the reaction performance needed to reveal the colorimetric output or to produce an efficient curing effect?

Response: Indeed, photoactivation efficiency is influenced by various factors including viscosity, polarity, molecular structure, and concentration. We've conducted additional experiments to further examine the kinetics of the photoreaction. By extending the temporal range, both P-EA and P-PA reached their plateaus of the restoration rates with 3 h UV irradiation (Fig. 3). All payloads exhibited an increase in viscosity (Supplementary Table 1) and a darkening in color (Fig. 2h and Supplementary Fig. 9c, 9f) upon exposure to the UV light, indicating restricted diffusion and diminished light penetration. The viscosity increases for both P-EA and P-PA were greater than that observed for P-HA under the same UV dose, leading to slower kinetics and lower restoration rates. To explore the concentration effect, we prepared three capsules containing 20 wt%, 50 wt%, and 80 wt% P-HA in EPA, respectively. The restoration efficiency increased when the payload

became less concentrated (Supplementary Fig. 11a) as dilution lowered viscosity and absorptivity of the solution, thus in turn facilitated diffusion and light penetration.

In this study, 80 wt% concentration was employed to achieve a balanced restoration rate and amine content. As high UV intensity facilitates photoactivation (Supplementary Fig. 11b) and the light-induced reactions typically slow down over time, P-HA capsules and P-EA capsules experienced 1 h exposure to a 365 nm UV light at an intensity of 30 mW/cm² are desired for damage-sensing and self-healing respectively.

We have updated Fig. 3 with extended kinetics data, provided Supplementary Fig. 11 with additional control experiments including concentration effect and UV irradiation at an intensity equivalent to the UV strength of sunlight, and expanded our discussion regarding concentration, UV intensity, light penetration, and viscosity in the Results section on pages 10 and 11.

"Given the efficient photoactivation, payload with 80 wt% protected-amines was selected for further studies to ensure an ample amine content, although the restoration efficiency further increased when the payload became less concentrated as dilution facilitated diffusion and light penetration (Supplementary Fig. 11a)."

"Nevertheless, a dramatic reduction of photoactivation efficiency was observed with a further decrease of UV intensity to 3 mW·cm⁻², a value corresponding to the sunlight UV strength (Air Mass 1.5 Spectra, ASTM G-173).⁴⁴ The restoration rate remained below 35% with an extended UV exposure for 3 h (Supplementary Fig. 11b)."

"The activation of amines further slowed down over time because of the increased payload viscosity (Supplementary Table 1) and decreased light transmission through the capsules as the photolysis continued (Supplementary Fig. 9)."

We've also outlined the recommended photoactivation conditions for P-HA and P-EA capsules in the Discussion section on page 17.

"Our initial screening studies revealed that the photolysis kinetics are highly dependent on the characteristics of the payload, suggesting that molecular architecture, polarity, and viscosity are essential for specific applications, particularly those requiring highly concentrated active amines and flowable payloads. Meticulous administration of UV light is essential for effective amine activation. For P-HA and P-EA capsules reported in this study, exposure to a 365 nm UV light at an intensity of 30 mW·cm² for 1 h ensures sufficient reactivity for damage-sensing and self-healing."

•Similar comments apply to the claim that the spherical geometry and high surface area enhance photolysis. The rationale behind the proposed controlled experiments is not entirely clear, particularly in terms of concentration considerations (bulk vs non-bulk). I suspect that reaction rates/efficiency may be influenced by concentration, which in turn affects viscosity and diffusion.

Response: The control experiment depicted in Supplementary Fig. 12 compares the photoactivation kinetics of the same 80 wt% P-HA in EPA solution stored in capsules versus in a quartz cuvette. We've added a theoretical analysis in the supplementary figure to illustrate that the power of light received per unit volume is approximately two orders of magnitude higher when the protected-amines are retained in the small-scale spherical structures. Concentrations of the

protected-amines were kept constant for these studies. Microcapsules significantly enhance irradiation efficiency by considerably increasing the specific surface area.

The reaction rate is indeed dependent on the concentration and viscosity of the solution, as discussed in our response to the above comment.

•The rationale behind the selection of capsules concentration for all demonstrators is not entirely clear to me. It would be beneficial to provide more insights into the motivation for these ratios.

Response: The influence of capsule concentration on the performance of damage-sensing was investigated in our previous studies (ACS Cent. Sci. 2016, 2, 598; Adv. Mater. 2016, 28, 2189). We showed that capsule loading as low as 2 wt% was sufficient to provide decent self-reporting capability, and polymer materials loaded with 10 wt% capsules demonstrated good mechanical properties. Since the NBD-Cl reporting system had both reactants stored in discrete compartments, we set the total capsule loading at 10 wt% to ensure good reporting and mechanical properties. The ratio of NBD-Cl and amine capsules was set at 3-to-1 (i.e. 7.5 wt% NBD-Cl capsule + 2.5 wt% amine capsule) so that both concentrations are above 2 wt% and the self-healing ability can potentially be integrated in our future studies by replacing the NBD-Cl/solvent capsule with the NBD-Cl/epoxy capsule.

For the colorimetric reporting epoxy, 2 wt% amine capsules were employed because BPB was dispersed uniformly in the polymer.

We have added the following descriptions in "damage-sensing" of the Results section on page 12.

"Capsule concentration was selected based on our previous studies to ensure damage-sensing ability while minimizing impact on the polymer properties."^{20,39}

•Furthermore, for the sake of reproducibility by peers, including an image of the photoprocess setup and providing additional details would be greatly helpful. The direction, distance, and intensity of light significantly influence the outcome of the photochemistry. I believe clarity in this regard is essential to avoid issues with reproducibility.

Response: We fully agree that UV irradiation setup is essential for photoactivation of the amines. We've added a schematic as Supplementary Fig. 19 and provided detailed descriptions in the Methods section on page 19 to facilitate reproduction of this research.

"As-prepared microcapsules containing protected aliphatic amines were irradiated in an Intelliray UV flood system (Uvitron International, MA) for up to 3 h (Supplementary Fig. 19). Amine capsules were powder-like, which facilitated a mono-layer distribution on a reflective aluminum tray. UV intensity was precisely controlled by adjusting power of the UV lamp and sample distance from the light source. Reflective aluminum foil was employed on the side walls to ensure 360° UV irradiation. The UV intensities at various locations were verified using a UV light meter."

•The authors present various reporting systems that differ in their chemistry, presumably due to differences in the matrix. Perhaps the authors could offer some insights or comments on this matter.

Response: We wanted to demonstrate the capabilities of the amine capsules in developing a range of damage-reporting polymers. We designed our studies to include diverse polymer matrices, fluorogenically and chromogenically indicating mechanisms, damage modes, and structural designs ranging from an amine-fluorophore dual-capsule system to a single amine capsule coupled with a dispersed dye layout.

The difference in the polymer matrices is indeed a crucial factor that we considered. NBD-Cl, along with numerous other dyes, exhibits poor solubility in PDMS, highlighting a material where amine capsules are critical. We prepared NBD-Cl microcapsules to showcase the dual-capsule scheme.

For epoxy, BPB was selected because it provided a chromogenic damage-reporting mechanism, and demonstrated a functionality that was previously unattainable.

We have added the following sentence into the Results section on page 12 to elaborate the design of NBD-Cl reporting system.

"The dual-capsule enables damage reporting in polymer matrices that are not compatible with certain indicators, such as the insoluble NBD-Cl in PDMS."

Details regarding the BPB reporting system have been included in the Discussion section on pages 16-17.

REVIEWERS' COMMENTS

Reviewer #1 (Remarks to the Author):

After the revision, the authors have answered all the comments and solved the issues. Thus, it is recommended for publication in NC without further modification.

Reviewer #2 (Remarks to the Author):

The manuscript was revised accurately answering all my questions. I appreciate bringing the readers up to speed specifically regarding the photolysis pathway of NPPOC and the previously investigated mechanical properties of the capsules. By the way, the synthesis from 2 to the protected amines can likely be done with carbonyldiimidazole or dimethylcarbonate as phosgene equivalents if the authors have safety concerns (rightfully so) about phosgene solutions. The manuscript is now in an excellent state for publication in Nature Communications.

Reviewer #3 (Remarks to the Author):

Revision manuscript NCOMMS-23-47591

Comments to the authors:

The revisions made to the manuscript titled in response to earlier comments have been thoroughly reviewed. It is acknowledged that all concerns raised have been adequately addressed, resulting in significant improvements to the manuscript.

The additions and modifications made to the presentation of methodologies have provided a clearer understanding of the photoreaction kinetics, addressing the previously noted lack of depth in this aspect. The incorporation of additional controls has also strengthened the validity of the findings, enhancing the robustness of the study. Based on the thorough revisions and improvements made, I believe that the manuscript now meets the standards expected for publication in Nature Communications.

I would like to thank the authors for their diligent efforts in addressing my previous comments and for their commitment to enhancing the manuscript. I look forward to seeing the work published and its contributions to the field.

Point-by-Point Response to Reviewer Comments

We extend our gratitude once more to the reviewers for their insightful and constructive comments during the peer-review process. At present, there are no additional comments necessitating further revision.

Response to Reviewer 1

After the revision, the authors have answered all the comments and solved the issues. Thus, it is recommended for publication in NC without further modification.

Response: We thank the reviewer for the positive evaluation of our work.

Response to Reviewer 2

The manuscript was revised accurately answering all my questions. I appreciate bringing the readers up to speed specifically regarding the photolysis pathway of NPPOC and the previously investigated mechanical properties of the capsules. By the way, the synthesis from 2 to the protected amines can likely be done with carbonyldiimidazole or dimethylcarbonate as phosgene equivalents if the authors have safety concerns (rightfully so) about phosgene solutions. The manuscript is now in an excellent state for publication in Nature Communications.

Response: We express our gratitude to the reviewer for their positive evaluation of our work. We greatly appreciate their suggestions regarding the alternative synthetic route, which we believe will greatly benefit our future studies.

Response to Reviewer 3

The revisions made to the manuscript titled in response to earlier comments have been thoroughly reviewed. It is acknowledged that all concerns raised have been adequately addressed, resulting in significant improvements to the manuscript.

The additions and modifications made to the presentation of methodologies have provided a clearer understanding of the photoreaction kinetics, addressing the previously noted lack of depth in this aspect. The incorporation of additional controls has also strengthened the validity of the findings, enhancing the robustness of the study. Based on the thorough revisions and improvements made, I believe that the manuscript now meets the standards expected for publication in Nature Communications.

I would like to thank the authors for their diligent efforts in addressing my previous comments and for their commitment to enhancing the manuscript. I look forward to seeing the work published and its contributions to the field.

Response: We sincerely appreciate the reviewer's support in the publication of our work.